# Sex-Specific Dysconnective Brain Injuries and Neuropsychiatric Conditions such as Autism Spectrum Disorder Caused by Group B *Streptococcus*-Induced Chorioamnionitis

**DOI:** 10.3390/ijms241814090

**Published:** 2023-09-14

**Authors:** Seline Vancolen, Taghreed Ayash, Marie-Julie Allard, Guillaume Sébire

**Affiliations:** 1Department of Pharmacology & Therapeutics, McGill University, Montreal, QC H3G 1Y6, Canada; seline.vancolen@mail.mcgill.ca; 2Department of Pediatrics, Research Institute of the McGill University Health Center, Montreal, QC H3G 1Y6, Canada

**Keywords:** androgen, autism spectrum disorder, behavioral deficits, brain injury, chorioamnionitis, fetal inflammatory response syndrome, IL-1, interleukin-1 receptor antagonist, polymorphonuclear cells, preterm birth

## Abstract

Global health efforts have increased against infectious diseases, but issues persist with pathogens like Group B Streptococcus (GBS). Preclinical studies have elaborated on the mechanistic process of GBS-induced chorioamnionitis and its impact on the fetal programming of chronic neuropsychiatric diseases. GBS inoculation in rodents demonstrated the following: (i) silent and self-limited placental infection, similar to human chorioamnionitis; (ii) placental expression of chemokines attracting polymorphonuclear (PMN) cells; (iii) in vitro cytokine production; (iv) PMN infiltration in the placenta (histologic hallmark of human chorioamnionitis), linked to neurobehavioral impairments like cerebral palsy and autism spectrum disorders (ASD); (v) upregulation of interleukin-1β (IL-1β) in the placenta and fetal blood, associated with higher ASD risk in humans; (vi) sex-specific effects, with higher IL-1β release and PMN recruitment in male placenta; (vii) male offspring exhibiting ASD-like traits, while female offspring displayed attention deficit and hyperactivity disorder (ADHD)-like traits; (viii) IL-1 and/or NF-kB blockade alleviate placental and fetal inflammation, as well as subsequent neurobehavioral impairments. These findings offer potential therapeutic avenues, including sex-adapted anti-inflammatory treatment (e.g., blocking IL-1; repurposing of FDA-approved IL-1 receptor antagonist (IL-1Ra) treatment). Blocking the IL-1 pathway offers therapeutic potential to alleviate chorioamnionitis-related disabilities, presenting an opportunity for a human phase II RCT that uses IL-1 blockade added to the classic antibiotic treatment of chorioamnionitis.

## 1. Background

Inflammation exerts a natural biological response against pathogens, thus acting as a double-edged sword, combining anti-infectious strikes with collateral damage. The noxious impact of infection results in great part from the inflammatory response generated by it. Recent epidemiological and preclinical studies provide a strong body of evidence supporting a relationship between Group B *Streptococcus* (GBS) and neurobehavioral disorders in the offspring [1,2,3,4].

Preclinical studies have shown dose-, time-, and sex-dependent effects of maternal immune activation (MIA) on intrauterine fetal demise, preterm birth, and unfavorable neurobehavioral outcomes in the offspring, such as autism spectrum disorders (ASD), cerebral palsy (CP), attention deficit/hyperactivity disorders (ADHD), and learning disabilities [5,6,7]. In the vast majority of these preclinical studies, pathogen-driven MIA was triggered by inactivated immunostimulants [5,6,8] such as lipopolysaccharide (LPS) from *Escherichia coli* acting mainly through toll-like receptor (TLR)-4, or polyinosinique-polycytidylique acid [poly (I:C)], a synthetic analog of viral ribonucleic acid (double-stranded RNA), acting through TLR-3 [9].

The nature of the pathogen modulates the profile of the MIA and thereby patterns of brain injury and neurobehavioral outcomes in the exposed offspring. Hence, it is important to conduct studies investigating the role of each pathogen commonly affecting pregnant mothers to specify their immune responses and the resulting neurobehavioral impairments. GBS is a common infection during human pregnancy, which only recently started to be studied preclinically with regard to the profile of GBS-induced MIA and its neurobehavioral impact on the offspring [10,11,12,13,14,15,16,17,18,19,20,21,22,23,24,25,26,27,28,29,30,31,32,33,34,35,36,37,38,39,40,41,42,43,44,45].

## 2. GBS Infection during Pregnancy

GBS is a common bacterium that asymptomatically colonizes the urogenital tract in 15–20% of pregnant women [8,10]. This bacterium ascends from the vaginal tract and cervix—or migrates via the hematogenous route—to the amniotic fluid to trigger inflammation of the placenta and fetal membranes, termed chorioamnionitis [11]. GBS-induced MIA can impact the timing of delivery by stimulating uterine contractions and triggering premature or preterm membrane rupture in humans, mice, but not rats (Figure 1) [7]. It also impacts the vulnerable brain of the developing fetus even without any bacterial translocation [5,6,10,26]. Thanks to the genital and anal GBS colonization screening tests for pregnant women recommended in most countries, antibiotic intrapartum prophylaxis can be administered at the time of delivery if the mother is GBS-positive between 35 and 37 weeks of pregnancy [12,13]. While this intrapartum antibiotic prophylaxis prevents neonatal infection by GBS in 88% of cases, it does not treat GBS-induced chorioamnionitis occurring well-before delivery [15], which is associated with a higher risk (odd ratio of 1.57) of perinatal death and postnatal morbidities when compared to placebo in humans [12,13]. 

## 3. GBS-Induced Maternal, Placental, and Fetal Immune Responses

There are several strains of GBS that are based on the structure of capsular polysaccharide. Serotype Ia and serotype III are the most prevalent, accounting for 25–50% and 10–25% of human perinatal infections, respectively [14,15]. GBS colonization can be transient, intermittent, or persistent throughout pregnancy. GBS uses various means of invasion and defense that allow it to become a natural component of vaginal microbiota [16]. Following contact between the bacterial and host cells, a cascade of events occurs simultaneously. On one hand, GBS triggers the innate immune responses induced either by pathogen-associated molecular patterns (PAMPs, such as lipoteichoic acid, lipoproteins, peptidoglycan, and β-hemolysin) engaging distinct TLR pathways, or by the β-hemolysin component acting on the inflammasome NOD-like receptor (NLR)-P3 pathway (Figure 1) [17,18,19,20]. On the other hand, the adaptive immune process leads to antibody opsonization, consisting of phagocytic cells (monocytes, PMN, or natural killer cells) expressing opsonin receptors (Fc receptor and complement receptor (CR)) to attract and eliminate coated GBS (Figure 1) [21]. This cascade of events stimulates the synthesis and release of several pro-inflammatory molecules, including cytokines (such as interferon (IFN), TNF-α, IL-6, IL-1β, IL-18) and chemokines (C-X-C), attracting PMN infiltration and activation, as well as prostaglandins and matrix metalloproteinases (MMPs) synthesis and release (Figure 1) [22]. This complex inflammatory response is important to control the infection, but it also generates deleterious placental and fetal effects.

## 4. Human Perinatal GBS Infection

GBS infection during pregnancy can result in urinary tract infection, chorioamnionitis, deciduitis, bacteremia, and/or sepsis [23]. GBS also induces fetal inflammatory response syndrome (FIRS) [24]. Fetal or newborn GBS infection can be very detrimental, generating complications such as meningitis and sepsis, which result in a heavy burden of unfavorable long-term outcomes [16,25]. This is because the bacteria may have either infected the fetus or may have remained within the placenta, but triggered FIRS. Two forms of neonatal GBS infections can manifest in the newborn: either early-onset disease or late-onset disease. The former typically occurs within the first seven days of life and can present with pneumonia, respiratory failure, and/or septicemia [16,25]; while GBS late-onset disease occurs in infants up to three months of age and presents with symptoms such as bacteremia, with a high risk of meningitis (50%) [16,24]. Altogether, GBS is a major threat; thus, further understanding its pathophysiology is crucial.

## 5. Maternal, Placental, and Fetal Inflammatory Changes in GBS-Exposed Placenta

Several studies have indicated that human cell types express a wide range of inflammatory chemokines, cytokines, and antimicrobial proteins, as well as release extracellular traps and undergo cell death in response to GBS exposure (Table 1) [26,27,28,29,30,31,32,33]. Patras et al. infected human epithelial cell types with different strains of GBS, including serotypes I, III, and V, and found that, depending on the strain, the bacteria displayed different abilities to adhere to and survive intracellularly [28]. Interestingly, GBS serotype V showed greater intracellular survival and less cytokine production compared to serotype Ia and III [28]. These differences may in part be explained by strain-specific changes in cellular signaling cascades, impacting downstream responses including phagocytosis/survival of GBS, cell death, and cytokine production [34]. However, specific inflammatory cytokines were universally induced in response to infection in cell types such as placental macrophages, trophoblasts, and endothelial and epithelial cells [30,31,32,33]. Strong and early IL-1β increase, as well as TNF-α and/or IL-6 increases, were documented following the exposure of ex vivo human choriodecidual tissues or cell lines to live or inactivated GBS [33,35]. Interestingly, uvaol, a component of olive oil, acting as a down regulator of NF-kB translocation, as well as IL-1Ra, dampened the GBS-induced human placental inflammation [31,33,36]. Macrophages play important roles in placental invasion, angiogenesis, and tissue remodeling and development, representing 20–30% of leukocytes in gestational tissues in humans [29,37]. Doster et al. focused on placental macrophages and found that, like neutrophils, they release extracellular traps and contain other placento-toxic proteins such as histones, myeloperoxidase, and neutrophil elastase [29].

Maternal, placental, and fetal inflammatory responses have been studied in various animal models of GBS-induced chorioamnionitis (Table 1) [5,6,7,8,38,39,40,41,42]. In most studies summarized in Table 1, several pro-inflammatory cytokines were upregulated in GBS models of chorioamnionitis. However, in one study by Andrade et al., three pro-inflammatory cytokines were significantly lower in the serum of infected pups compared to that of the uninfected ones [40]. This decrease could be attributed to the distinct characteristics of the developing immune system in neonates [40]. Overall, IL-1β plays a key role in the placental immune response against GBS infection. It drives placental PMN infiltration and FIRS-induced neurodevelopmental impairments [36]. Increased levels of IL-1β were detected in maternal and fetal serum from urogenital GBS-colonized mothers, which have been associated with early human term deliveries (between 37 and 39 weeks) [43]. IL-1 blockade provides a protective effect against GBS-induced chorioamnionitis and subsequent neurobehavioral impairments in the rat offspring [46].

Sex differences exist in perinatal inflammatory processes (Table 1). Significantly higher levels of IL-1β, cytokine-induced neutrophil chemoattractant-1 (CINC-1/CXCL1), and PMNs infiltration were found in inactivated GBS-exposed male, compared to female, maternofetal tissues in rodents [8,41]. Androgens in males upregulated the placental innate immune response in the GBS-induced chorioamnionitis rat model [44]. It would be interesting to further investigate the effects of androgens on the interactions between GBS and macrophages or PMN to understand through which innate immune mechanism androgens exert their modulation of the innate immune system. 

On another note, preclinical studies have shed light on MIA triggered by other pathogen components, including lipopolysaccharide (LPS) from *E. Coli.* Girard et al. and others have reported that systemic end-gestational LPS infection in dams causes a significant increase in placental cytokine levels, followed by brain injuries, and results in high fetal mortality [45,46,47]. It has also been demonstrated that LPS from *E. coli* triggers preterm birth in mice; however, LPS-exposed rat dams deliver on term [48,49,50].

Hence, rodents are useful models for comparing the effects of pathogen-induced MIA, their perinatal and long-term impacts, and their prevention, but a lot of work remains to be conducted to fully understand the molecular mechanisms at play in several anatomical compartments (placenta, fetal blood, brain) and their effect on the various outcomes.

## 6. Brain Injuries Associated with GBS-Induced Inflammation

MIA can disrupt neurodevelopmental events shaping the immature brain and result in life-long brain injury. Perinatal activation of the immune system and altered profiles of circulating inflammatory molecules have been associated with recognizable morphological patterns of injuries in the offspring’s brain in preclinical models (Table 2) [6,8,38,40,51]. These brain injuries might be the consequences of either direct or indirect impairments of end-gestational neurodevelopmental processes such as oligodendrocyte, astrocyte and microglial differentiations, neuronal network construction, and potentially other mechanisms that are not yet well understood [52,53].

As can be seen in Table 2, Andrade et al. reported an increase in activated microglia following in utero GBS III infection in a preclinical model [40]. Post mortem histological brain studies of ASD patients revealed increased expression of microglia-specific markers in the prefrontal cortex compared to matched controls [54].

In general, white matter injury (WMI) is the most common type of brain injury in human preterm newborns [55]. This type of brain injury, which is linked to cognitive and behavioral anomalies, manifests in humans through subtle modifications in the WM microenvironment. These alterations include WM atrophy (thinning of the corpus callosum, ventriculomegaly, due to tissue loss and shrinkage of the brain parenchyma adjacent to the ventricles), and other subtle patterns of dysmyelination due to impaired oligodendrocyte maturation [55].

Similar to WMI in humans, a reduced thickness of the corpus callosum was found in the male rat offspring exposed to GBS in utero (Table 2) [6,8,38]. In a rat model of end-gestational exposure to inactivated GBS, it was shown that periventricular WM was affected in the offspring [6]. This WMI was characterized by decreased mature and non-proliferative oligodendrocytes (CC-1)-positive cells, without any difference in oligodendrocyte transcription factor 2 (Olig2)-positive stained cells (all oligodendrocytes) (Table 2) [6]. These results suggest that GBS-induced inflammation might skew the oligodendrocyte maturation processes without much cell loss. Other rodent models of diffuse WMI previously reported that WM microstructure alterations are detectable by diffusion tensor imaging for several months, and even longer—i.e., adulthood in human—following inflammatory insults, and are associated with impaired cognitive abilities [56].

Beyond GBS-induced MIA, fetal brain injuries have been studied using different infectious causes of chorioamnionitis, as well as different rodent species. For instance, rat models of LPS-induced chorioamnionitis show the activation of the maternal pro-inflammatory cytokine profile [57,58,59]. Consequently, pups display severe brain damage including WM brain lesions, as well as a significant increase in microglial cells in the forebrain [59,60]. These patterns of dysconnective brain injuries are relatively similar to those associated with GBS-induced chorioamnionitis [59].

## 7. Sex-Dichotomic Behavioral Impairments Due to Exposure to GBS-Induced Chorioamnionitis

The studies profiling the neurobehavioral impact of GBS-induced maternofetal inflammation were summarized in Table 3 [6,8,38,40,51,61]. GBS-exposed male offspring display deficits in social interaction, communication, processing of sensory information, and preference toward maternal cues [6,8,38,40,61]. These reported sex-specific behavioral impairments are interesting considering the higher susceptibility of the human male population for neurobehavioral disorders such as ASD. Such neurobehavioral anomalies closely mimic behavioral characteristics of human ASD [62,63]. Accordingly, in an epidemiological study by Limperopoulos et al., 76% of human preterm newborns positive for ASD screening had a history of chorioamnionitis [64]. Multiple preclinical and clinical studies have also displayed a link between perinatal infection and inflammation, preterm birth, and subsequent brain damage, contributing to other motor and psychiatric disorders such as CP, schizophrenia, and ADHD [65,66]. Recently, a nationwide cohort study using Danish and Dutch registry data to study infants with a history of GBS disease suggested that boys were at higher risk of neurodevelopmental impairments [67]. Overall, preclinical and clinical studies both concur in supporting the key role of sex-dichotomic effects of MIA on behavioral outcomes in the offspring.

## 8. Anatomo-Behavioral Correlations in GBS-Exposed Offspring

Changes at the levels of neural structure and function have behavioral implications. Decreases in volumes of periventricular WM, including the corpus callosum and external capsule, were found in the rat offspring exposed to GBS (Table 2). There was also a reduced density of microglia Iba-1-positive cells in the corpus callosum of male rat offspring [38]. Depletion of microglia interferes with brain wiring processes and related myelination in rodents [68,69,70]. Similarly, post mortem studies on ASD individuals revealed dysconnectivity in WM structures such as the corpus callosum [71]. Furthermore, a 16–24% reduced thickness of the corpus callosum was observed in male and female rat offspring in utero exposed to inactivated GBS [6]. Since these structures are largely involved in sensory integration, it is not surprising that these rat offspring show impaired olfaction (nest-seeking) and startle response to auditory stimuli (decreased pre-pulse inhibition) [38]. In the same line, ASD patients present impairment in sensory integration and modulation.

Magnetic resonance imaging (MRI) and in situ analysis revealed a significant enlargement of the lateral ventricles in male rat offspring following in utero exposure to formaldehyde-killed GBS (Table 3). Data from a large multi-site MRI dataset reveal asymmetry of the hippocampus and lateral ventricles in ASD individuals compared to non-ASD patients in general [72]. Enlarged lateral ventricles are also characteristic of WMI, which is the most common type of brain injury found in preterm infants who are at high risk of developing ASD (OR: 16), especially in the context of human chorioamnionitis [64].

In addition, as shown after in utero exposure to inactivated GBS in a preclinical model, the fronto-temporal circuits, located within the abnormally thinner external capsule adjacent to the lateral ventricles, likely contribute to their enlargement [6]. Notably, these anomalies of the external capsule are relevant to ASD because such fronto-temporal connections play a key role in regulating behaviors that are affected in ASD manifestations, such as anxiety, sensory integration, and others. Finally, thinner primary motor cortices were detected in GBS III-exposed males, but not females (Table 3), and correlated with the severity of CP-like traits in rats [8]. This is relevant to the unbalanced sex ratio towards males in human CP [8].

## 9. Translating Placento- and Neuro-Protective Research into Clinical Practice

The identification of TLR2/6 and β-hemolysin/(NLR)-P3 pathways, as well as IL-1, as key mediators in the inflammatory response triggered by GBS-induced sepsis has prompted clinical trials of anti-inflammatory interventions to protect maternofetal organs. In preclinical models of chorioamnionitis triggered by GBS and LPS, the IL-1 blockade has already demonstrated placenta- and feto-protective effects [36,73]. Of particular interest, a Phase I/IIa study of the drug Anakinra, which is an IL-1Ra analogue, has been underway since February 2022 [74]. This drug presents a potential strategy for preventing perinatal inflammation in premature infants, which is associated with morbidities such bronchopulmonary dysplasia, pulmonary hypertension, and cerebral diffuse WMI [75]. Briefly, enrolled infants born between 24 weeks 0 days (24^0^) and 27^6^ will receive Anakinra over the first 21 days of birth, and the frequency of adverse outcomes/events will be monitored [74,75]. A systematic review analyzing randomized control trials, observational studies, and case reports shows that IL-1 blockers are safe during human pregnancy with no significant increase in adverse outcomes [75]. Therefore, IL-1 blockade represents a promising approach to protect the placenta, improve pregnancy outcomes, and reduce the risk of GBS-induced unfavorable neurological outcomes in humans [46]. While the benefits of IL-1 blockers are evident, it is crucial to acknowledge that their effectiveness hinges on the early-stage diagnosis of chorioamnionitis. Presently, GBS screening is exclusively conducted between 35 and 37 weeks of pregnancy. Such biomarkers will allow for optimal treatment with antibiotics combined with anti-inflammatory medications, adapted for each patient according to the infectious and/or sterile trigger(s).

## Figures and Tables

**Figure 1 ijms-24-14090-f001:**
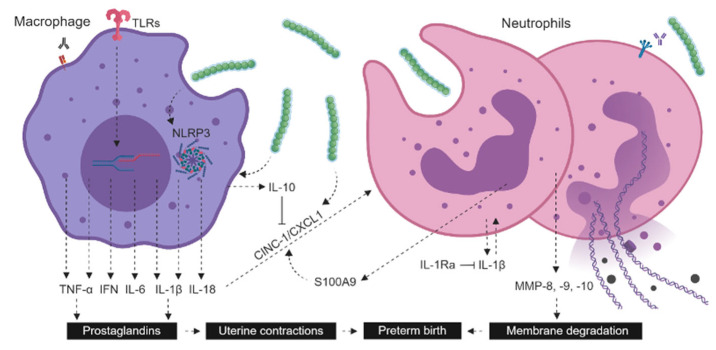
Key: GBS-induced inflammatory signaling pathways. Created with BioRender.com.

**Table 1 ijms-24-14090-t001:** Profile of cytokines, chemokines, and other antimicrobial proteins produced in vitro or in vivo, and associated cellular and/or tissular changes in response to GBS exposure.

References	Species	Immunogen, Dose, Route	Timing of Administration	Maternal, Placental, and Fetal Changes in the GBS-Exposed Rodents	
				Males	Females
Bergeron et al., 2013 [6]	Lewis rats	Killed β-hemolytic GBS Ia 16955, 10^10^ CFU, ip	Every 12 h G19 to G22	↑ cavitary lesions and Malpighian metaplasia in placenta (G22); ↑ PMN in decidua, junctional zone of placenta (G22); no difference in CD68+ or Iba-1+ macrophages in labyrinth of placenta (G22)	
Randis et al., 2014 [7]	C57BL6/J mice	Live β-hemolytic GBS V NCTC10/84, 10^7^ CFU, intravaginal	G13	50% positive maternal blood cultures (G17); 88% positive placental cultures for GBS (G17), 43% positive fetal blood cultures (G17); ↑ GBS infiltrates in decidua and labyrinth of placenta (G17); ↑ pathology score	
Bergeron et al., 2016 [5]	Lewis rats	Killed β-hemolytic GBS Ia, strain 16955 10^10^ CFU, ip	Every 12 h G19 to G22	↑ IL-1β in maternal serum (3, 24, 48, 72 h); ↑ PMN (24, 48, 72 h), ↑ MMP-8 mRNA, ↑ MMP-10 mRNA, ↑ S100A9 mRNA, ↑ UPA mRNA (6 h), ↑ CXCL1 (3 h), ↑ MMP-10 (48 h) and IL-1β (72 h) in placenta; ↑ IL-1β (72 h) in fetal serum	Not studied
Allard et al., 2017 [38]	Lewis rats	Live β-hemolytic GBS Ia, 16955, 10^9−10^ CFU, ip	G19	↑ GBS infiltrates in placenta (G22); ↑ PMN in decidua, junctional zone, and labyrinth (G22)	↑ GBS infiltrates in placenta (G22); ↑ PMN in decidua and junctional zone, but not labyrinth (G22)
Kothary et al., 2017 [39]	C57BL6/J mice	Live β-hemolytic GBS V, 10^3^ CFU, intravaginal	G13	GBS invasion in vagina, uterus, placenta, decidua, and fetus (G15); ↑ CD45+ Neu7/4+ GR1+ neutrophil cells in placenta and decidua (G15); ↑ lactoferrin HDAB within fetal placental tissue (G15); ↑ NETs (G15)	
Andrade et al., 2018 [40]	BALB/c mice	Live β-hemolytic GBS III BM110, 3 × 10^4^ CFU, intravaginal	G17 and G18	↓ TNF-α, ↓ IL-17, and ↓ IFN-γ in pups’ serum (P5). No difference KC, MIP-1α, IL-1β, IL-6, and IL-10 in pups’ serum (P5). No difference KC, MIP-1α, IL-1β, IL-6, TNF-α, IL-17, IFN-γ, and IL-10 in pups’ brain (P5)	Not studied
Allard et al., 2019 [8]	Lewis rats	Killed β-hemolytic GBS III BM110, 10^10^ CFU, ip	Every 12 h G19 to G22	↑ GBS infiltrates in placenta (G22); ↑ PMN infiltrates in decidua, junctional zone, and labyrinth (G22)	
Allard et al., 2019 [41]	Lewis rats	Live β-hemolytic GBS Ia 16955, 10^9−10^ CFU, ip	G19	↑ IL-1β (48, 72 h), ↑ IL-6 (48, 72 h), ↑ TNF-α (48, 72 h), ↑ IL-10 (72 h), and ↑ CXCL1 (48, 72 h) in maternal serum	
				↑ GBS infiltrates in placenta (48 h, 72 h); ↑ PMN in decidua (48, 72 h), junctional zone (48, 72 h), and labyrinth (72 h); ↑ CXCL1 (72 h), ↑ S100A9 (48, 72 h), ↑ MMP-8 (72 h), ↑ IL-1β (48, 72 h), ↑ IL-6 (48, 72 h), ↑ TNF-α (48, 72 h), and ↑ IL-10 (72 h) in placenta; ↑ IL-1β (72 h) and TNF-α (72 h) in fetal serum	↑ GBS infiltrates in placenta (48 h, 72 h); ↑ PMN in decidua (48, 72 h) and junctional zone (48, 72 h), but not labyrinth of placenta; ↑ S100A9 (48, 72 h), ↑ MMP-8 (72 h), and ↑ IL-1β (48 h, 72 h); ↑ IL-6 (48, 72 h), ↑ TNF-α (48 h, 72 h), and ↑ IL-10 (72 h) in placenta; ↑ TNF-α (72 h) but not IL-1β in fetal serum

Abbreviations: CD, cluster of differentiation; CFU, colony-forming units; CXCL, Chemokine (C-X-C) ligand family; G, gestational day; GBS, Group B *Streptococcus*; h, hour; IFN, interferon; IL, interleukin; ip, intraperitoneal; KC, keratinocyte chemoattractant; MIP, macrophage inflammatory proteins; MMP, matrix metalloproteinase; NETs, neutrophils elaborate extracellular traps; P, postnatal day; PMN, polymorphonuclear cell; S100A9, S100 calcium-binding protein A-9; TNF, tumor necrosis factor; UPA, urokinase plasminogen activator.

**Table 2 ijms-24-14090-t002:** Histopathological changes in the brain of offspring in utero exposed to GBS-induced chorioamnionitis in rodent models.

References	Species	Immunogen, Dose, Route	Timing of Administration	Morphological Changes in the GBS-Exposed Offspring’s Brains	
				Males	Females
Barichello et al., 2013 [51]	Wistar rats	Live GBS III 10^6^ CFU/mL, intracerebral	P3–P4	↓ BDNF levels in the prefrontal cortex (P70); ↓ BDNF levels in the hippocampus (P70); and ↓ NGF levels in the hippocampus (P70)	
Bergeron et al., 2013 [6]	Lewis rats	Killed β-hemolytic GBS Ia 16955, 10^10^ CFU, ip	Every 12 h G19 to G22	↑ area of lateral ventricles (P40); ↓ thickness of CC (P40); no difference Iba-1 in CC (P40)	↓ Iba-1 in CC (P40); ↓MBP in CC (P40)
				↓ thickness of EC (P40); ↓ CC-1 in CC (P40); no difference Olig2 in CC (P40); no difference GFAP in CC (P40)	
Allard et al., 2017 [38]	Lewis rats	Live β-hemolytic GBS Ia 16955, 10^9−10^ CFU, ip	G19	↑ area of LV (P40); ↓ thickness of CC (P40); and ↓ thickness of EC (P40)	No difference area of LV (P40); no difference thickness of corpus callosum and EC (P40)
Andrade et al., 2018 [40]	BALB/c mice	Live β-hemolytic GBS II BM110, 3 × 10^4^ CFU, intravaginal	G17 and G18	↑ Evans blue leakage (BBB permeability) (P5), ↓ thickness of PC (P5), ↑area of LV (P5), ↑ TUNEL in MC, striatum, PC and hippocampus (P5), ↑ GFAP in hippocampus (CA3 region) (P5), ↑ activated microglia in hippocampus (CA3 region) (P5), no difference thickness of EC (P5)	Not studied
Allard et al., 2019 [8]	Lewis rats	Killed β-hemolytic GBS III, BM110, 10^10^ CFU/mL, ip	Every 12 h G19 to G22	↓ thickness of CC (P40); ↓ thickness of primary MC (P40); ↓ Iba-1 in CC (P40)	No difference thickness of CC and primary MC (P40); no difference Iba-1 in CC (P40)
				↓ MBP in corpus CC (P40); no difference GFAP in CC and primary MC (P40), and Iba-1 in primary MC (P40)	

Abbreviations: BBB, Blood-brain barrier; BDNF, Brain-derived neurotrophic factor; CC, corpus callosum; CFU, colony-forming units; EC, external capsule; Iba-1, Ionized calcium-binding adapter molecule 1; G, gestational day; GBS, Group B *Streptococcus*; GFAP, Glial fibrillary acidic protein; h, hour; ip, intraperitoneal; LV, lateral ventricle; MBP, Myelin basic protein; MC, motor cortex, NGF, nerve growth factor; P, postnatal day; PC, parietal cortex.

**Table 3 ijms-24-14090-t003:** Sex-dichotomic behavioral impairments due to exposure to GBS-induced chorioamnionitis.

References	Species	Immunogen, Dose, Route	Time of Administration	Behavioral Changes in the GBS-Exposed Offspring	
				Males	Females
Barichello et al., 2013 [51]	Wistar rats	Live GBS III, 10^6^ CFU, intracerebral	P3–P4	No difference in motor, exploratory activity, habituation memory in the OF (P70) ↓ aversive memory compared with the long-term memory test in the Step-down inhibitory avoidance task (P70)	
Bergeron et al., 2013 [6]	Lewis rats	Killed β-hemolytic GBS Ia 16955, 10^10^ CFU, ip	Every 12 h G19 to G22	↑ latency to reach familiar odor (P9) ↓ locomotion in OF (P15–25) ↓ latency to fall of rotarod (P30–40) ↓ PPI to acoustic stimuli (P35) ↓ number and duration of social interactions (P40)	No difference to reach familiar odor (P9) No difference in the OF (P15–25) No difference of latency to fall of rotarod (P30–40) No difference for PPI (P35) ↑ duration of social interactions (P40)
Allard et al., 2017 [38]	Lewis rats	Live β-hemolytic GBS Ia 16955, 10^9−10^ CFU, ip	G19	↓ USVs (P7) ↑ latency to reach familiar odor (P9) ↑ locomotion in the OF (P20) ↓ duration of social interactions (P40)	No difference for USVs (P7) No difference to reach familiar odor (P9) No difference in the OF (P15–25) No difference in social interactions (P40)
				↓ PPI to acoustic stimuli (P35)	
Allard et al., 2018 [61]	Lewis rats	Killed β-hemolytic GBS Ia A909, 10^10^ CFU, ip	Every 12 h G19 to G22	No difference in the OF (P15–25) No difference for the latency to fall of rotarod (P30–40) No difference in the EPM (P35–40) No difference in the EPM (P105–110)	No difference in the OF (P15–25) ↓ latency to fall of rotarod (P40) No difference in the EPM (P35–40) ↑ open maze exploration; and ↑ distance in the EPM (P105–110)
Andrade et al., 2018 [40]	BALB/c mice	Live β-hemolytic GBS III BM110, 3 × 10^4^ CFU, intravaginal	G17 and G18	↓ distance; ↓ time spent in central area; ↓ rearing; ↓ exploration in the OF (P90); and ↓ working memory in the Radial Maze (P90)	Not studied
Allard et al., 2019 [8]	Lewis rats	Killed β-hemolytic GBS III BM110, 10^10^ CFU, ip	Every 12 h G19 to G22	↓ distance and mobility in the OF (P25)	No difference in the OF (P15–25)
				↓ startle response to acoustic stimuli (P35–65)	

Abbreviations: CFU, colony-forming units; G, gestational day; GBS, Group B *Streptococcus*; EPM, elevated plus-maze; h, hour; ip, intraperitoneal; OF, open field; P, postnatal day; PPI, prepulse inhibition; USV, ultrasonic vocalization.

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
