# Peer review of "Sex-Specific Dysconnective Brain Injuries and Neuropsychiatric Conditions such as Autism Spectrum Disorder Caused by Group B Streptococcus-Induced Chorioamnionitis"

_ijms, 2023, doi:10.3390/ijms241814090_

Round 1
Reviewer 1 Report
This narrative review summarizes the sex-specific neurological sequalae of GBS induced chorioamnionitis. The manuscript is well structured and scientifically sound. Overall, the topic is very relevant, and deserves to be published. Some comments:
1. General:
1.1. The authors jump between preclinical and clinical studies and at times make it difficult for the reader to follow. Please ensure that the reader is able to follow the authors’ thought train at all times. The authors use this review as a motivation for a human phase II RCT. If this is the goal, then all arguments should lead to this point.
2. Specific comments
2.1. Line 68: “induced MIA can impact”
2.2. Line 72: “intrapartum antibiotic prophylaxis”
2.3. Line 112: In pregnancy the endometrium undergoes decidualization and therefore “deciduitis” should be named and not endometritis.
2.4. Line 152. Please define what the authors mean by “early human term deliveries”.
2.5. Table 1. The reader does not know what the last row of this table means. Is this a summary or part of the last study mentioned. Please edit table.
2.6. Line 200. As a whole this paragraph is difficult to read. The authors start of this paragraph talking about WMIs. But it is not clear to the reader if this is in context of GBS or not. Then the authors jump between preclinical and clinical data. Please edit the intro into this paragraph so that it is easy to read.
2.7. Line 234. This paragraph on briefly summarises the preclinical data in one sentence and then moves on to clinical observational data. If sex differentiation is one of the main arguments in this review, then the authors should state their opinion on these studies. Comment, critique or appraise the preclinical data. Is it really rigorous enough for a translational study?
2.8. Table 3. Please define ip.
2.9. Line 269 – 270. Is this in context of GBS or ASD patients in general?
2.10. Line 289. Please define the study of Anakinra clearly. What is the intervention, what is the primary outcome, in short define the methodology. Then comment on it to motivate why another study needs to be set up.
2.11. Line 298. In the last sentence the authors motive for the development of new biomarkers. This is a different aim as stated in the introduction. Please clarify.
See comments to authors.
Author Response
Our answers to the reviewer:
“1.1. The authors jump between preclinical and clinical studies and at times make it difficult for the reader to follow. Please ensure that the reader is able to follow the authors’ thought train at all times. The authors use this review as a motivation for a human phase II RCT. If this is the goal, then all arguments should lead to this point.”
The authors have made adjustments to clarify the distinction between animal and human findings in the text.
“2.1. “induced MIA can impact””
We have corrected this error (Line 67).
“2.2. “intrapartum antibiotic prophylaxis””
We have corrected this error (Lines 72-73).
“2.3. In pregnancy the endometrium undergoes decidualization and therefore “deciduitis” should be named and not endometritis.”
We have replaced the word endometritis with deciduitis (Line 110).
“2.4. Please define what the authors mean by “early human term deliveries”.”
Early human term deliveries are defined as between 37 and 39 weeks, this has been clarified in the text (Lines 150-151).
“2.5. Table 1. The reader does not know what the last row of this table means. Is this a summary or part of the last study mentioned. Please edit table.”
This row was not correctly merged, it is the same reference. We have corrected this error (Table 1).
“2.6. As a whole this paragraph is difficult to read. The authors start of this paragraph talking about WMIs. But it is not clear to the reader if this is in context of GBS or not. Then the authors jump between preclinical and clinical data. Please edit the intro into this paragraph so that it is easy to read.”
To ensure clarity, this paragraph has been split into two sections, the first relating to WMIs in general and the second to GBS-induced WMI (Lines 199-214).
“2.7. This paragraph on briefly summarises the preclinical data in one sentence and then moves on to clinical observational data. If sex differentiation is one of the main arguments in this review, then the authors should state their opinion on these studies. Comment, critique or appraise the preclinical data. Is it really rigorous enough for a translational study?”
The authors agree and have added additional detail to this paragraph, providing opinions on the mentioned studies (Lines 234-251).
“2.8. Table 3. Please define ip.”
This has now been defined in the abbreviations (Table 3).
“2.9. Is this in context of GBS or ASD patients in general?”
ASD patients in general, this has been added to the text (Lines 274-276).
“2.10. Please define the study of Anakinra clearly. What is the intervention, what is the primary outcome, in short define the methodology. Then comment on it to motivate why another study needs to be set up.”
The authors agree that the study must be outlined clearly and have therefore added additional detail (Lines 296-300).
“2.11. In the last sentence the authors motive for the development of new biomarkers. This is a different aim as stated in the introduction. Please clarify.”
These aims are interconnected. While we emphasize the potential advantages of IL-1 blockers, it is crucial to recognize that their effectiveness hinges on the early-stage diagnosis of CA. No screening test for GBS is currently performed at earlier gestational age (i.e. before 35-37 weeks of pregnancy), and there is no diagnostic test available to diagnose histological CA in utero (it becomes only possible after birth based on the pathological examination of the placenta), hence the need to identify diagnostic biomarkers of CA. Authors have added additional information to this paragraph to clarify (Lines 305-310).
Reviewer 2 Report
The manuscript is overall well written and well presented.
However, when the Authors present the experimental data about GBS in pregnancy, placenta, etc... it is not always clear when they refer to animal or humans. I would suggest revising that part making a clear distinction between animal and human findings (in vivo or post-mortem).
Author Response
“The manuscript is overall well written and well presented. However, when the Authors present the experimental data about GBS in pregnancy, placenta, etc... it is not always clear when they refer to animal or humans. I would suggest revising that part making a clear distinction between animal and human findings (in vivo or post-mortem).”
The authors thank the reviewer for the positive feedback and have made adjustments to clarify the distinction between animal and human findings in the text.
Round 2
Reviewer 1 Report
The authors have sufficiently addressed al concerns.
Author Response
Dear Editor,
As requested, we are glad to re-submit a revised version of our manuscript.
We are very grateful to you and the reviewers for the good comments which improved the quality of our manuscript.
Please find hereunder our point-by-point replies to the reviewers and editors comments’.
“In table 1, reference “Andrade et al 2018 (32)” (doi: 10.1038/s41467-018-05492-y): The results regarding IL1B are wrongly stated. Please revise the mentioned article and appropriately correct the information. Please fully revise the rest of the information for possible additional mistakes.”
We thank you for bringing this to our attention, it has been edited (Table 1).
“Lines 147-149: The authors state that, as summarized in Table 1, several pro-inflammatory cytokines are upregulated in GBS models of chorioamnionitis. However, in Table 1, in the reference by Andrade 2018, the levels of various inflammatory cytokines were significantly lower in the serum of infected pups compared to uninfected pups. Please refer to this result in the text and provide some information regarding the explanation to why a lower concentration of these inflammatory cytokines was found in the serum of infected pups.”
An additional statement covering this has been added (Lines 148-151).
“The references are incorrectly numbered, especially in the Tables. For instance: Number 85 and 8 is the same citation but using a different format. In the Tables, reference number 32 is used to refer both Allard 2017 (Table 2) as well as Andrade 2018 (Table 1). Please correct. Also, reference numbers 32 and 33 are both used to refer to Andrade 2018 elsewhere in tables; however, in the reference list, these numbers are Vogel 2014 and Larsen 2008, respectively. Please, fully revise all references for accuracy and formatting.”
These references have been adjusted.
“Lines 197-200: In this paragraph, the information from the preclinical (mouse) and human studies referred in the two sentences is not connected, even if the link in the provided information is clear and relevant. Please help the reader by providing a small phrase relating the results from both studies.
The same suggestion as in point number 4, for the paragraph in lines 279-282. This would help in understanding the relevance of the preclinical studies for understanding the clinical observations in patients.”
This has been added, we have also ensured that the clarification between clinical or pre-clinical work is clear throughout the rest of the paper.
With our best regards,
Guillaume Sébire, MD, PhD